# Detection of Nail Oncometabolite SAICAR in Oral Cancer Patients and Its Molecular Interactions with PKM2 Enzyme

**DOI:** 10.3390/ijerph182111225

**Published:** 2021-10-26

**Authors:** Rushikesh Patel, Ajay Kumar Raj, Kiran Bharat Lokhande, Mazen A. Almasri, Khalid J. Alzahrani, Asma Saleh Almeslet, K. Venkateswara Swamy, Gargi S. Sarode, Sachin C. Sarode, Shankargouda Patil, Nilesh Kumar Sharma

**Affiliations:** 1Cancer and Translational Research Lab, Dr. D. Y. Patil Biotechnology & Bioinformatics Institute, Dr. D. Y. Patil Vidyapeeth, Pune 411033, Maharashtra, India; rushikeshpatel143@gmail.com (R.P.); aveer9680@gmail.com (A.K.R.); 2Bioinformatics Research Laboratory, Dr. D. Y. Patil Biotechnology & Bioinformatics Institute, Dr. D. Y. Patil Vidyapeeth, Pune 411033, Maharashtra, India; kiran.lokhande@dpu.edu.in; 3Oral and Maxillofacial Surgery Department, King Abdulaziz University, Jeddah City 21589, Saudi Arabia; malmasri@kau.edu.sa; 4Department of Clinical Laboratories Sciences, College of Applied Medical Sciences, Taif University, P.O. Box 11099, Taif 21944, Saudi Arabia; Ak.jamaan@tu.edu.sa; 5Oral and Maxillofacial Surgery and Diagnostic Sciences Department, Riyadh Elm University, Riyadh 12611, Saudi Arabia; Asma.Almeslet@riyadh.edu.sa; 6MIT-School of Bioengineering Sciences & Research, MIT-Art, Design and Technology University, Pune 412201, Maharashtra, India; venkateswara.swamy@mituniversity.edu.in; 7Department of Oral Pathology and Microbiology, Dr. D. Y. Patil Dental College and Hospital, Dr. D. Y. Patil Vidyapeeth, Pimpri, Pune 411033, Maharashtra, India; gargi14@gmail.com; 8Department of Maxillofacial Surgery and Diagnostic Sciences, Division of Oral Pathology, College of Dentistry, Jazan University, Jazan 45142, Saudi Arabia; dr.ravipatil@gmail.com

**Keywords:** oral cancer, metabolite biomarkers, SAICAR, pyruvate kinase M2, in silico studies

## Abstract

Oncometabolites are known to drive metabolic adaptations in oral cancer. Several oncometabolites are known to be shared between cancer cells and non-cancer cells including microbiotas to modulate the tumor microenvironment. Among potential oncometabolites, succinylaminoimidazolecarboxamide ribose5′-phosphate (SAICAR) supports the growth and invasiveness of cancer cells by pyruvate kinase M2 (PKM2) enzyme in a glucose starved tumor microenvironment. There is a significant gap that shows the detection of SAICAR in biological samples including nails of oral cancer patients. Metabolite identification of SAICAR was investigated in the nails of oral cancer patients using novel vertical tube gel electrophoresis (VTGE) and LC-HRMS. Further molecular docking and molecular dynamics simulations (MDS) were employed to determine the nature of molecular interactions of SAICAR (CHEBI ID:18319) with PKM2 (PDB ID: 4G1N). Molecular docking of SAICAR (CHEBI ID:18319) was performed against pyruvate kinase M2 (PDB ID: 4G1N). Data suggest the presence of oncometabolite SAICAR in nails of oral cancer. Molecular docking of SAICAR with PKM2 showed appreciable binding affinity (−8.0 kcal/mol) with residues including ASP407, THR405, GLU410, ARG443, GLY321, ARG436, HIS439, LYS266, and TYR466. Furthermore, MDS confirmed the specific binding of SAICAR within the activator site of PKM2 and the stability of SAICAR and PKM2 molecular interactions. In conclusion, SAICAR is a promising oncometabolite biomarker present in the nails of oral cancer patients. A significant activation potential of SAICAR exists with the PKM2 enzyme.

## 1. Introduction

Tumor heterogeneity is contributed by various components including cellular and non-cellular components such as oncometabolite [1,2]. Mutual exchanges of various metabolites are an important feature among cancer cells and non-cancer cells including immune cells, stromal cells, and microbiotas in the niche of the oral tumor microenvironment (3–4). Well-established evidence on metabolic heterogeneity is a contributing factor in various oral tumor hallmarks including invasiveness, metastasis, and drug resistance [1,2,3,4,5]. 

The importance of oncometabolite in various types of cancers, including oral cancer, is widely appreciated in the basic understanding of metabolic heterogeneity and thus is envisaged as a potential source of biomarkers [5,6,7]. Recently, novel molecular approaches were reported on the profiling of oncometabolites in biological samples of oral cancer and precancerous patients [7,8,9]. 

A key oncoprotein pyruvate kinase M (PKM) has a vital role in the metabolic adaptations of cancer cells. In the mammalian system, PKM has four types of isomers including protein kinase M PKM2. PKM2 is reported as a catalyst that converts phosphoenolpyruvate into pyruvate along with ATP and is associated with high expression in cancer cells [10,11,12,13].

In cancer cells, PKM2 works as a dimer to produce pyruvate. Several studies have revealed that oncometabolites namely succinylaminoimidazolecarboxamide ribose-5′-phosphate (SAICAR), fructose-1,6-P2, and serine can lead to the formation of tetramer from a dimer of PKM2 [10,11,12,13,14,15]. SAICAR, an intermediate product of the de novo purine nucleotide synthesis pathway, acts as an oncometabolite that supports the growth of cancer cells in a nutrient-limited medium [15,16,17,18,19,20]. 

Altered metabolism and survival under stressed conditions are some of the hallmarks of cancer. During metabolic reprogramming, cancer cells and cancer supporting stroma including microbiome establish a source and link partnership for the exchange of key oncometabolites including SAICAR, for growth and proliferation [21,22,23,24,25,26,27,28,29]. SAICAR is known as an important intermediate metabolite that is catalyzed by SAICAR synthetase in both cancer cells and the microbiome [30,31,32,33]. 

Based on the existing knowledge, there is a significant gap in the detection of SAICAR in biological samples including nails of oral cancer patients. Furthermore, molecular docking and molecular dynamics (MD) simulation data on SAICAR interactions with PKM2 are needed for better understanding and future mimetic of SAICAR as an anticancer drug. Hence, we report on the detection of SAICAR in the nail of oral cancer patients by using our novel and validated vertical tube gel electrophoresis (VTGE) methodology and molecular docking and MD simulations to predict the molecular interactions between SAICAR and PKM2. 

## 2. Materials and Methods

### 2.1. Study Population

OSCC (n = 5) and healthy controls (n = 6) were recruited at Dr. D. Y. Patil Dental College and Hospital, Pune, India. The participating subjects in both OSCC and healthy subjects were in the age group of 30–60 years. Before the commencement of this study, Institutional Ethics Committee approval was obtained (Ref. No. DYPV/EC I 14{2019 Date: 15 March 2019). In this study, participants were detailed on the purpose of the study, and informed consent was collected.

### 2.2. VTGE Assisted Purification of Nail Metabolites 

A novel vertical tube gel electrophoresis (VTGE) system was used to detect SAICAR in the nail metabolite lysate of oral cancer patients (Appendix A). A total of 80 mg of fingernail clippings of OSCC patients were dissolved in 800 µL of nail lysis buffer (5M Urea, 2.6 M thiourea, Tris-HCl (20 mM, pH-8.5) and beta-mercaptoethanol). Further, nail lysates were purified with the help of the VTGE metabolite purification system [34,35]. Purified nail metabolites of OSCC patients were detected by using LC-HRMS. During LC, the RPC18 column (Zorbax, 2.1 × 50 mm, 1.8 µm) was employed. Then metabolites were submitted to MS Q-TOF Quadrupole time-of-flight mass spectrometry (Q-TOF-MS) with positive electrospray ionization (ESI) M-H mode.

### 2.3. Molecular Docking 

SAICAR was detected as a potential oncometabolite in the nails of OSCC patients, so we have proceeded to in silico studies on SAICAR. To perform molecular docking, potential oncometabolite SAICAR (CHEBI ID:18319) was retrieved as a ligand from the ChEBI database in SDF format. Then OpenBabel software was employed for the conversion of ligands in SDF format into PDB format. Energy minimization of ligands before performing molecular docking is an important step to obtain stable conformation of the ligand. Avogadro software was used for energy minimization of the ligand by selecting the steepest descent method and MMFF94s force field [36]. Pyruvate kinase M2 (PDB ID: 4G1N) was considered as the target receptor protein. It was downloaded in PDB format from the Protein Data Bank (PDB). To free all the binding pockets on the receptor, bound ligands were removed by deleting heat atoms from the PDB file. Then the protein PDB file was opened in AutoDock Tool 4.2. To perform the steps of protein preparation, the steps consist of the removal of water molecules, bond correction, assigning AD4 type atoms, the addition of polar hydrogens, and Kollman charges [37] to the receptor. AutoDock Vina Software was used for molecular docking of SAICAR oncometabolite with PKM2 protein [7].

AutoDock Vina has inbuilt automatic grid maps. First, blind docking was performed to confirm the active binding sites of the ligand. Blind docking includes the covering of the whole receptor protein with a grid box of appropriate size. The docking includes organized conformational expansion of the ligand and further interaction of oncometabolite to the active site residues of the receptor occurs. Visualization of the binding position of SAICAR into the binding pockets of the PKM2 was performed using a discovery studio visualizer.

### 2.4. Molecular Dynamics (MD) Simulations

Desmond software was used for 10ns molecular dynamics (MD) simulation of the oncometabolite SAICAR—PKM2 complex to confirm the binding stability and strength of the complex (39). This software comes with the features of adding pressure, temperature, volume system, and many other functions to complete the protein–ligand binding. Protein–Ligand complex was immersed in a water-filled orthorhombic box of 10 Å spacing (40). The SAICAR–PKM2 complex system is solvated by 21,066 water molecules using an extended three-point water model (TIP3P) with periodic boundary conditions. These studies were performed with a run of 10ns and temperature 300 K, considering certain parameters such as integrator as MD. The conformational changes upon binding of SAICAR with PKM2 were recorded with the help of 1000 trajectories frames generated during 10ns MD simulation. The root mean square deviation (RMSD) was calculated to confirm the deviation in the conformation of ligand and protein.

### 2.5. vNN-ADMET Toxicity Prediction

The canonical smiles of selected oncometabolite SAICAR were downloaded from the PubChem database. Further, these chemical structures were tested in vNN-ADMET web servers (41) to predict the substrate specificity of P-gp, mutagenicity (AMES mutagenesis), and cytotoxicity.

## 3. Results

### 3.1. Identification of Oncometabolite from Nails of Oral Cancer Patients

Previous findings on oral cancer and precancerous lesions, the contribution of oncometabolite including SAICAR is appreciated [23,24,25,28]. However, preclinical, clinical, and molecular interaction attributes of SAICAR in the case of oral cancer are highly limited. Additionally, the importance of SAICAR has relevance as a biomarker and a source of future anticancer drugs as metabolic mimetics that can block the pro-cancer enzymes. In line with existing views, the authors report on a novel approach to the detection of SAICAR in the nails of oral cancer patients with the assistance of a novel and in-house vertical tube gel electrophoresis (VTGE) tool [26,29] (27).

Here, elutes of nail metabolites prepared by VTGE were identified by using LC-HRMS in a positive ESI mode. By the analysis of the LC-HRMS profile, SAICAR is detected as an oncometabolite in the nails of oral cancer patients compared to the healthy control. SAICAR showed distinctive mass ion spectra with the chemical formula (C13 H19 N4 O12 P), *m*/*z* (436.0626), and mass (454.0726) in a positive ESI mode (Figure 1) (see Appendix B). This is the first study that showed the presence of SAICAR in the nail of oral cancer patients. This observation prompted the investigation of the biological relevance of SAICAR in cancer metabolic adaptations by using molecular docking and molecular dynamics (MD) simulations. 

### 3.2. Molecular Docking 

It is understood that SAICAR binds to the dimeric form of PKM2 and converts it into the trimetric form [15,16]. The progression of the tumor is associated with the activation of inactive PKM2 by SAICAR and other oncometabolite [17]. There is a lack of molecular interaction studies between SAICAR to PKM2 that leads to the activation of this enzyme. 

Here, the authors have performed molecular docking to understand the binding pattern of SAICAR upon PKM2. 

Molecular docking of SAICAR with PKM2 (PDB ID: 4G1N) predicted appreciable binding affinity (−8.0 kcal/mol) (Figure 2A). Upon the detailed visualization of interactive amino residues, SAICAR binds through 13 polar bonds to the binding residues including ASP407, THR405, GLU410, ARG443, GLY321, ARG436, HIS439, LYS266, and TYR466 within the active pocket of PKM2 (Figure 2B–D). 

### 3.3. Molecular Dynamics (MD) Simulations

MD simulations were performed for the duration of 10ns to examine the stability of the ligand–protein complex. The root mean square deviation (RMSD) of ligand and protein was calculated during the 10ns of the simulation period. RMSD was aimed to measure the average change in the displacement of C-α atoms for 1000 frames concerning a reference frame (initial docked conformation). The RMSD plot of PKM2 protein indicates that displacement of protein is up to 2.5 Å, which is acceptable within range (1–3 Å) for the stability of protein throughout the simulation period. The RMSD plot of the SAICAR–PKM2 illustrates that initial displacement in the conformation of the complex up to 4ns, afterward it maintains the equilibration state (Figure 3).

The left axis depicts the RMSD plot for the PKM2 enzyme regarding changes in structural conformation during 10ns simulation time. The changes in the protein RMSD value are observed in the range of 1–3 Å. The right-Y axis denotes the RMSD value of bound SAICAR (CHEBI ID:18319) to PKM2 and this value is not significantly larger than the RMSD value of the PKM2 enzyme (PDB ID: 4G1N). Thus, the RMSD plot describes the stable ligand–protein complex.

MD simulation study also comprises the graphical presentation of protein–ligand contacts, categorized by the type of bonds. It explains the availability of interaction between ligand and protein throughout the simulation. This graph shows that SER406 and ASP407 have hydrogen bonds for 60% to 90% of simulation time (Figure 4A). Another graph for protein–ligand interaction depicts the interaction of each residue with ligand in each time frame of the simulation. There is a darker shade of orange in the graph, which describes that some residues have more than one specific contact with the ligand. Simulation data suggested that ASP407, GLU285, SER 406, ASP407, and GLU410 have more than one bond with the ligand. MD simulations also provide the schematic diagram of detailed ligand interaction with residues of the protein, it shows the interaction, which generates for more than 30% of the simulation time of the selected 0.00 to 10.00 ns trajectory (Figure 4B). Amino acid residues such as ASP407, THR405, SER406, and ARG443 were shown to interact with ligands in a schematic diagram with the nature of the residues in different colors. Altogether, data collected from molecular docking and MD simulations suggested strong evidence of SAICAR–PKM2 interactions.

### 3.4. vNN-ADMET Toxicity Prediction

The relevance of SAICAR produced within the tumor microenvironment by cancer cells and cancer supporting cells including the microbiome is predicted in the perspective of the extent of the toxicity and suitable substrate of drug transporters. vNN-ADMET hinted that even if SAICAR is produced in large quantities within the tumor microenvironment, no cellular toxicity and adverse effects are predicted (Figure 5). Interestingly, SAICAR appears to be a good substrate of P-gp drug transporter. 

## 4. Discussion

PKM2 is described as one of the limiting enzymes in glycolysis, which induces the formation of pyruvate and ATP from phosphoenolpyruvate (PEP) and ADP in proliferating cancer cells [27,30]. PKM2 is present in the dimeric form with a higher *K*m value for the phosphoenolpyruvate (PEP) substrate, accordingly the dimeric form of PKM2 is in an inactive state at a normal physiological state [31,32].

There are reports that SAICAR concentration in cancer cells increases gradually during the starvation of glucose and eventually stimulates PKM2 for cancer progression [15,16]. PKM2 expression promotes the uptake of glucose upon inhibition of oxygen and expression of low-active PKM2 dimer brings the accumulation of metabolites such as serine, phosphoenolpyruvate (PEP), and glucose-6-phosphate [33,34]. Serine binds to PKM2 and activates it by maximizing the use of glucose and in glucose-deprived conditions switches to SAICAR for activation of dimeric PKM2 independent of FBP [7,35,36]. Furthermore, the clinical relevance of the PKM2 enzyme is linked with oral cancer and supports the growth and proliferation during abnormal glucose metabolism [1,38,39]. 

Besides existing views, there is a complete gap in the clinical relevance of SAICAR as an oncometabolite in OSCC. In the present work, the detection of SAICAR in the nails of oral cancer patients is attributed to a novel VTGE assisted approach that helped in a clear detection of SAICAR. On the other hand, SAICAR is not detectable in healthy control.

SAICAR is detected as an oncometabolite and there is evidence of the overexpression of the PKM2 enzyme in oral cancer. Therefore, we attempted to reveal the molecular interaction of SAICAR with the PKM2 enzyme by using molecular docking and MDS. As well as the strong binding affinity of SAICAR, interactive amino acid residues including ASP407, THR405, GLU410, ARG443, GLY321, ARG436, HIS439, LYS266, and TYR466 of PKM2 the presence of a good number of 13 polar bonds. Apart from molecular docking data, the MD simulation study of the SAICAR–PKM2 complex confirmed the key amino acid residues ASP407, THR405, SER406, and ARG443. In literature, THR405, SER406, and ASP407 residues are spanning within the key pocket of allosteric activator sites in PKM2 enzyme for activator oncometabolites such as F16BP and serine [33,40]. Therefore, present molecular interaction studies provide additional information on the activation binding sites of SAICAR upon PKM2 and support the existing in vitro and in vivo evidence on SAICAR as an activator of PKM2.

### Sharing of Saicar between Oral Cancer Cell and Microbiome

Recently, the role of SAICAR was highlighted in the activation of PKM2 that promotes the growth and proliferation of cancer cells including oral cancer cells. (16) However, the source of SAICAR within the tumor microenvironment is cancer cells and the microbiome in the niche of the tumor. Interestingly, the generation of SAICAR with the help of SAICAR synthetase in bacterial cells is a well-known metabolic adaptation feature during glucose starvation. (15) Hence, a proposition is warranted to understand the possibility of the exchange of SAICAR produced by microbiomes such as *P. gingivalis*, *F. nucleatum*, *P. intermedia*, *E. coli*, and *S. aureus* to cancer cells for the activation of key metabolic enzyme PKM2. Sharing of oncometabolite resources between cancer cells and the microbiome is a potential interaction that supports the link between microbiome and cancer cells.

Prediction studies from webserver to predict absorption, distribution, metabolism, excretion, and toxicity (vNN-ADMET) hinted at the suitability of SAICAR as a good substrate of P-gp, which is known as a key play in drug resistance and other environmental stress. This supported our proposition on the possibilities of exchanges of SAICAR among cancer cells, microbiome, and non-cancer cells such as macrophages to fuel up the metabolic adaptations during growth, proliferation, and drug resistance. Such prediction raises the possibility of metabolic exchange SAICAR among the cellular components within the tumor microenvironment. This prediction is well supported by the existing literature on the role of SAICAR and PKM2 activation is associated with the pro-inflammatory act by macrophages that may work as cancer-supporting non-cancer cells [41,42]. Finally a proposed model for SAICAR as a metabolite biomarker in the nails of oral cancer patients with overexpression of PKM2 and dietary patterns of oral cancer patients is presented in Figure 6.

## 5. Conclusions

In conclusion, we suggest the potential of SAICAR as oncometabolite biomarkers in the nails of oral cancer patients. Furthermore, SAICAR shows effective allosteric binding upon the PKM2 enzyme. SAICAR is predicted as a key oncometabolite that is shared among the various cellular components including the microbiome within the oral tumor microenvironment. Moreover, this paper highlights the relevance of in-house developed novel VTGE methodology that assists in the metabolic study including qualitative and quantitative in nature. This approach has prospects for uses in other biological samples including urine, serum and saliva with slight modifications. The authors propose that VTGE approach may have relevance various other cancer types other than oral cancer and metabolic diseases such as inborn errors and diabetes. This study may be helpful for future therapeutic approaches as mimetic of SAICAR that can disrupt the SAICAR–PKM2 interaction and this may disturb the metabolic landscape of cancer cells.

## Figures and Tables

**Figure 1 ijerph-18-11225-f001:**
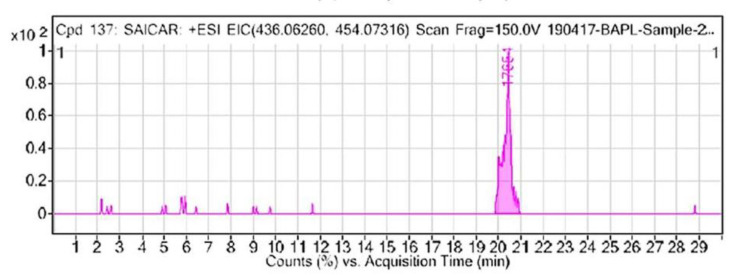
An oncometabolite SAICAR is detected in nails of oral cancer patients. A positive ESI extracted ion chromatogram (EIC) of SAICAR was detected during LC-HRMS of nail lysates purified with the help of a novel VTGE tool.

**Figure 2 ijerph-18-11225-f002:**
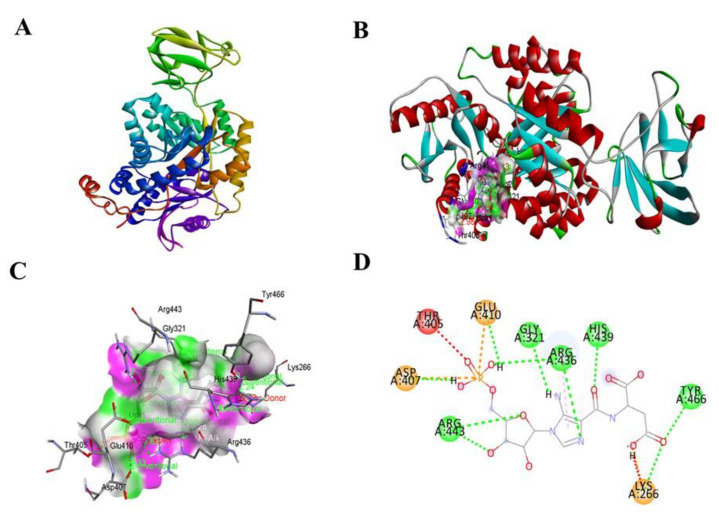
An oncometabolite SAICAR (CHEBI ID: 18319) shows strong molecular binding with pyruvate kinase M2 (PKM2) enzyme (PDB ID: 4G1N). Molecular docking and interaction of SAICAR with PKM2 were visualized with the help of Discovery Studio Visualizer. (**A**) Molecular docking affinity estimated by AutoDock Vina. (**B**) Three-dimensional view of interaction between SAICAR and PKM2 with binding residues, bond distances, and types of bonds. (**C**) The docked molecular structure between SAICAR and PKM2 is visualized in a 3D image depicting H-bond interactions (acceptor in green and donor in pink color) (**D**) Two-dimensional image of docked molecular structure between SAICAR and PKM2 derived from Discovery Studio Visualizer.

**Figure 3 ijerph-18-11225-f003:**
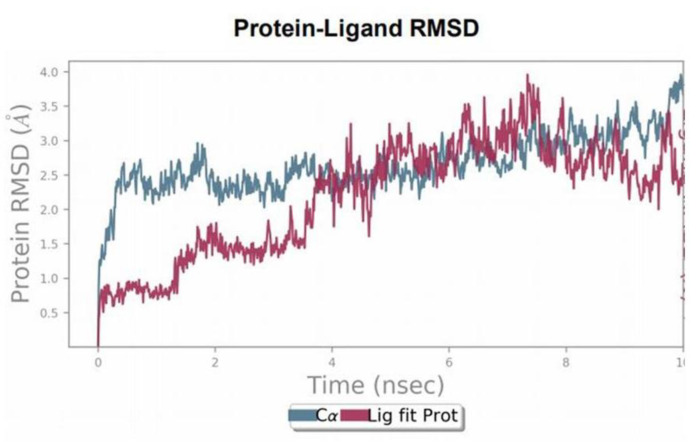
An oncometabolite SAICAR displays stable complex with PKM2. PKM2–SAICAR root mean square deviation (RMSD) plot for 10ns of time frame showing the stability of the complex between SAICAR depicts strong and specific binding to PKM2.

**Figure 4 ijerph-18-11225-f004:**
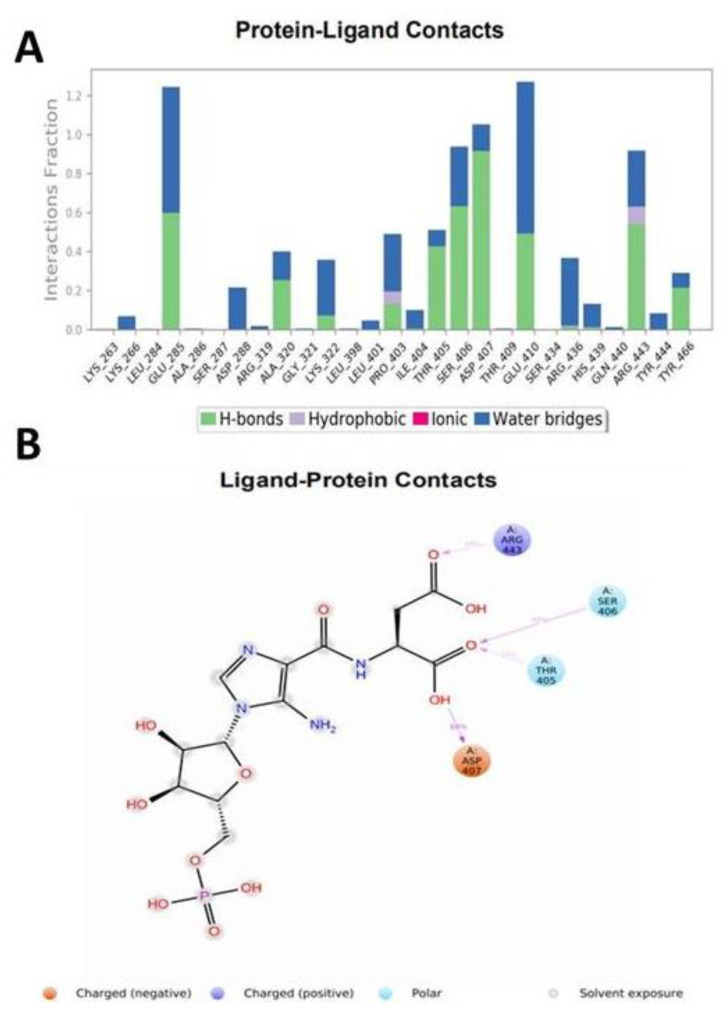
SAICAR shows specific contacts within the activation site of PKM2. (**A**) PKM2 and SAICAR interaction plot on the interaction between amino acid residues and ligand remained during 10 ns simulation. On the Y-axis, the interaction fraction shows the time of established interaction between key amino acid and ligand through different types of bonds such as hydrogen bonds, hydrophobic, ionic, and water bridges. (**B**) Schematic diagram on the interaction of the ligand with amino acid residues, which has remained for more than 30% of interaction time of the simulation. Here, various color combinations are used to represent the extent and nature of ligands to enzyme atomic interactions including ionic, hydrophobic, polar, water, and solvent exposure.

**Figure 5 ijerph-18-11225-f005:**
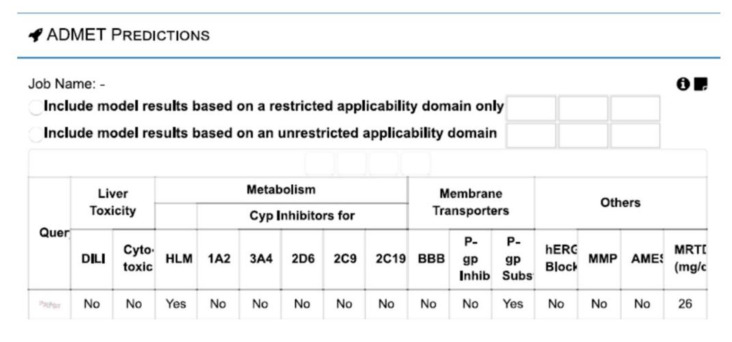
vNN-ADMET predicts SAICAR as a substrate of drug transporter P-gp and no adverse cell toxicity.

**Figure 6 ijerph-18-11225-f006:**
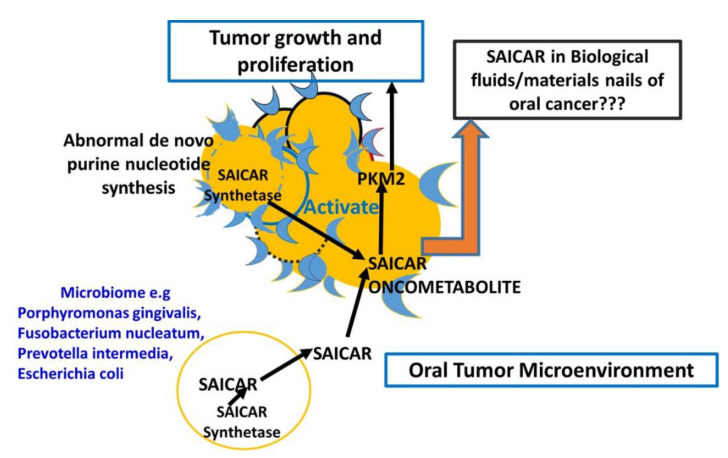
A proposed model on the role of SAICAR oncometabolite as a nail metabolite biomarker and predicted as a key metabolite in growth and proliferation by binding to PKM2.

## Data Availability

The data is available with the corresponding authors and can be made available on request.

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
