# Peer review of "Detection of Nail Oncometabolite SAICAR in Oral Cancer Patients and Its Molecular Interactions with PKM2 Enzyme"

_ijerph, 2021, doi:10.3390/ijerph182111225_

Round 1

Reviewer 1 Report

The authors have presented a well organized manuscript that documents the measurement of SAICAR in OSSC fingernails and a model of PKM2 interaction. The novelty it the technique and proposed model provide notable insight that is deserving of publication in this journey after addressing the following comments:

  • in the methods of sample measurements, please note if there is a specific weight of nails that is used per sample.
  • For figure 1 , please present these plots for all measured samples. Since there arent that many samples, it'd be nice to see the level of variation among OSSC and compared to controls.

Author Response

Point 1: In the methods of sample measurements, please note if there isa specific weight of nails that is used per sample.

Response 1: The authors appreciate the comment. For the preparation of nail lysates, 80 mg of fingernail clippings of OSCC patients and healthy controls were dissolved in 800 µl of nail lysis buffer (5M Urea, 2.6M thiourea, Tris-HCl (20mM, pH-8.5) and beta-mercaptoethanol). The same has mentioned in the revised manuscript at page 2 last paragraph.

Point 2: For figure 1, please present these plots for all measuredsamples. Since there aren’t that many samples, it'd be nice tosee the level of variation among OSSC and compared tocontrols.

Response 2: The authors appreciate the suggestion. In this method, after the use of novel VTGE assisted purification of nail lysates for the desired metabolites, we performed qualitative analysis of distinctive metabolites for both OSCC and healthy control. Later, we performed quantitative analysis of OSCC and healthy control for SAICAR and other metabolites. We found the acceptable level of SAICAR in OSCC and conversely, SAICAR was not detected in healthy control. The authors agree to the view that detailed quantitative measurement of SAICAR will enhance the validity and importance of this paper.

Reviewer 2 Report

Interesting biological study. Before publishing it some correction are needed.

Abstract it is essential that all the acronym are clearly defined when introduced for the first time in a text. Your study is of course full of acronym so this could be difficult and reductory for text readability, however, both in abstract and text, the crucial acronym of SAICAR should be described.

Materials and Methods

The protocol number of the EC approval must be referred 

Discussion The entire sub-section "SHARING OF SAICAR BETWEEN ORAL CANCER CELL AND MICROBIOME" must be reduced. It is out of the main focus of the paper and its relevance in the whole development of the study does not deverve a so great analysis.

Moreover it should be more widely emphasized the relevance of the VTGE assessing method.

Again: what is vNN-ADMET? please write it clearly before the acronym

Conclusion

Also in the conclusions the relevance and novelty of the VTGE is ignored, please focus it and its relevance from an eventual clinical utilization

Author Response

Point 1: Interesting biological study. Before publishing it some correctionare needed.

Response 1: We thank reviewer for their positive remark on our study.

Point 2:Abstract it is essential that all the acronym are clearly definedwhen introduced for the first time in a text. Your study is ofcourse full of acronym so this could be difficult and reductory fortext readability, however, both in abstract and text, the crucialacronym of SAICAR should be described.

Response 2: The authors have included and addressed the suggestions and appropriate changes are made in the revised manuscript including modifications for metabolites succinylaminoimidazolecarboxamide ribose5’-phosphate (SAICAR).

Materials and Methods

Point 3: The protocol number of the EC approval must be referred

Response 3: The authors have included the EC approval reference number (Ref.No.DYPV/EC I 14{2019 Date: 15 March 2019)

Discussion

Point 4:The entire sub-section "SHARING OF SAICARBETWEEN ORAL CANCER CELL AND MICROBIOME" must bereduced. It is out of the main focus of the paper and its relevancein the whole development of the study does not deserve a sogreat analysis.Moreover it should be more widely emphasized the relevance ofthe VTGE assessing method.

Response 4: We thank reviewer for this valuable input. As per the suggestion, efforts have been made to reduce and rewrite the section to make it more compatible with the context of the paper.

Point 5:Again: what is vNN-ADMET? please write it clearly before theacronym.

Response 5: The authors have included expanded form of vNN-ADMET as “webserver is a publicly available online platform to predict absorption, distribution, metabolism, excretion, and toxicity (vNN-ADMET)”

Conclusion

Point 6:Also in the conclusions the relevance and novelty of the VTGE isignored, please focus it and its relevance from an eventualclinical utilization.

Response 6: The authors have added additional points in the conclusion. “This paper highlights the relevance of in-house developed novel VTGE methodology that assists in the metabolic study including qualitative and quantitative in nature. This approach has prospects for uses in other biological samples including urine, serum and saliva with slight modifications. The authors propose that VTGE approach may have relevance various other cancer types other than oral cancer and metabolic diseases such as inborn errors and diabetes.
